# The limiting zero dynamics of discrete-time system based on forward triangle sample-and-hold

**Minghui Ou**[1,2]*, **Mingkun Ou**[3], **Haiyang Wang**[1]

**1** School of Big Data and Internet of Things, Chongqing Vocational Institute of Engineering, Chongqing, China, **2** College of Automation, Chongqing University, Chongqing, China, **3** Huahai Engineering Co., Ltd, China Railway Engineering Corporation, Shanghai, China

* ouminghui@cqvie.edu.cn

**Data Availability Statement:** All the data of the system model in this paper can be deduced from the numerical example based on the theorems of this paper. And all the simulation figures can be realized by MATLAB.

## Abstract

Zero dynamics have crucial effect on system analysis and controller design. In the control analysis process, system performance is influenced by the unstable zero dynamics, greatly. This study concerns with the properties of limiting zero dynamics when the signal of controlled continuous-time systems was reconstructed by forward triangle sample-and-hold (FTSH). FTSH is a newly sample and hold method in the signal reconstruction area. However, more theoretical details about the limiting zero dynamics of the resulting discrete-time systems need to be revealed. Firstly, the framework for the limiting zero dynamics in the situation of sufficiently small or large sample period is introduced. Furthermore, this study provides the stable conditions of limiting zeros in the two different sampling situations. The results indicate that one can select a suitable variable parameter value of FTSH to replace the sampling zeros of discrete-time system locate inside the stable region. This paper also reveals the truth that FTSH has the outstanding advantage compared with the BTSH using theoretical analysis method. Finally, example simulations verify the effectiveness of the results in this study.

## Introduction

In practical control engineering area, the continuous-time system can not be used to design and realize the control strategy, directly. We should deduce a corresponding discrete-time system to represent the original continuous-time system by using a sample and hold device. The zero properties of the continuous-time linear system affect the behavior of the controlled system. Moreover, for discrete-time system, the minimum phase (MP) can be reflected by the location of sampling zero, and it plays an important role in system analysis and control strategy design process [1, 2]. In the controller design process of system, the appearance of unstable zero can be seen as a great unfriendly obstacle to construct control strategy, such as model reference adaptive controller and model matching system [3]. Therefore, it is a meaningful work to avoid the unstable zero arise from the discretization process. A considerable amount of researchers have poured many efforts to study this question in recent decades [4–14].

**Funding:** This research was funded by the Science and Technology Research Program of Chongqing Municipal Education Commission (Grant No. KJZD-M202203401, KJQN202103401, KJQN202103413) and the Natural Science Foundation of Chongqing, China, (Grant No. cstc2021jcyj-msxmX0532, cstc2021ycjh-bgzxm0088), Program for Innovation Research Groups at Institutions of Higher Education in Chongqing (Grant No. CXQT21032), the University's Scientific Research Program of Chongqing Vocational Institute of Engineering (Grant No. 2022KJA03, JG222024). The funders had no role in study design, data collection and analysis, decision to publish, or preparation of the manuscript.

**Competing interests:** The authors have declared that no competing interests exist.

Discrete-time model of the continuous-time system relay on the reconstruction method about the input and output signals. During the process of the discretization, the relationship of the pole has a single mapping: $p_i \leftrightarrow e^{p_i T}$, $p_i$ is the continuous-time part and $e^{p_i}T$ is the discrete-time part, $T$ represents sampling period. However, this simple relation cannot be preserved for the zeros, more complicated transformations about zeros in the discretization process need to be concerned. The dynamics of discrete-time system is complex, only for the classification about the zeros have two categories: intrinsic zeros and limiting (or sampling) zeros, respectively [4, 15]. The properties of limiting zeros mainly depend on the method of discretization, the scale of sampling rate and the relative degree of the original system. Limiting zeros have no continuous-time counterpart and simple map relation in the $z$-domain. Many previous literature have revealed that the limiting zeros converge, as the sampling rate goes to infinity, to the roots of special polynomials (such as Euler-Fröbenious polynomials [4, 6, 16] and its' modified form [17]). Because of the truth that the limiting zeros is the function about sampling period $T$, only select some special case of sampling period one can deduce the expression of limiting zeros. For example, the sufficiently small or large sampling period attracts a lot of researches in [6, 10, 17–20]. Åström et al. provide the groundbreaking work to research the exiting question in the limit case [4]. Hagiwara et al. focus on the properties of limiting zeros when using zero order hold (ZOH) or first order hold (FOH) cascade into the control framework with sufficiently large or small sampling rate [6]. In [18], authors extended this work into fractional order hold (FROH). The stability of limiting zeros of discrete-time system in the case of backward triangle sample and hold (BTSH) for sufficiently small or large sampling period was discussed in [9]. These previous literature has revealed the truth that to research the properties of zeros with limit sampling rate value is an important work to explore more potential value about one specific sample and hold method, which will provide a theoretical reference in the practical engineering applications.

Two new sample-and-hold functions: forward triangle sample and hold (FTSH) and BTSH were investigated in [21, 22], recently. Two triangular sample and hold functions were considered as an alternative to the ZOH and shown that they can replace the sampling zeros of the corresponding sampled-data systems locate in unite circle by using BTSH or FTSH while ZOH can not to do. However, the mainly results in [21] lay emphasis on numerical investigation. Therefore, it is natural that one should analytical investigate the properties of zeros of a discrete-time system with the above two sample and hold functions. Consequently, Ou et al. [9, 23] were analytically investigated the properties of zeros and zero dynamics of sampled-data systems with BTSH for linear and nonlinear systems, respectively. They were also researched the sampling zero properties of discrete-time system with FTSH in [22], but only the case of sufficiently small sampling period was studied. Moreover, points to the fact that the properties of limiting zeros should be considered completely in all the case of sufficiently limiting sampling periods [4, 6]. It is a significant research work to explore the capabilities of sampling zero and advantages of FTSH with limiting sampling period.

The main purpose of this study is systematically to reveal the properties of limiting zeros of discrete-time system with FTSH in details. The present study clarifies the corresponding results of limiting zeros for sufficiently small and large sampling period when the FTSH is applied to discretize the continuous-time control system. Further, we also present the stable conditions of the limiting zeros.

The contributions of current study are marked as shown in the following three aspects:

(I) FTSH was used as the signal reconstruction, authors deduced a novel polynomial. The stable conditions of limiting zeros were provided, which is a vital complement to the results in [22].

(II) This paper analytically reveals the truth that FTSH has the outstanding advantage compared with the BTSH, which is coinciding with numerical research in [21].

(III) The theoretical framework of properties of limiting zeros for some eligible systems with FTSH was provided. Authors verified that by using the results in this study can replace the limiting zero in the stable region.

## Preliminaries

As mentioned in the introduction, mainly purpose of this paper is to reveal the sampling zeros, for FTSH discretization, under limiting value of the sampling period (i.e. $T \to 0$ and $T \to \infty$). For purpose of better understand the corresponding results of this paper, some well-known preliminaries knowledge are reviewed in this section.

The transfer function of a continue-time system is defined as $G_c(s)$. $G_F(z)$ represents the corresponding discrete-time transfer function of $G_c(s)$ under FTSH as the signal reconstruction, where $s$ denotes a complex variable for Laplace transform and $z = e^{Ts}$.

$$G_c(s) = \frac{b_m s^m + \cdots + b_0}{s^n + a_{n-1}s^{n-1} + \cdots + a_0}(b_m \neq 0), \tag{1}$$

the transfer function (1) is irreducible. The integer $r = n - m$ is the relative degree (i.e. pole excess). Then, one can obtain the state space expression about the controllable, observable and single-input single-output linear continuous-time system.

$$\dot{x}(t) = A_c x(t) + bu(t), \quad y(t) = cx(t), \tag{2}$$

where $A_c \in R^{n \times n}$, $b \in R^{n \times 1}$ and $c \in R^{1 \times n}$ represent $n$-th order state matrix, column and row vectors, respectively. Moreover, $u(t) \in R$, $x(t) \in R^{n \times 1}$ and $y(t) \in R$ represent the input, state vectors and output of the controlled system, respectively. The following formula provide a bridge between transfer function and state space expression.

$$G_c(s) = c(sI - A_c)^{-1}b. \tag{3}$$

Based on the different expression of linear system (1) and (2), those system zeros (also can be defined as transmission zeros or invariant zeros, those three names are coincide) can be deduced by computing the part of numerator polynomial in (1).

In this study, the discrete-time system model is composed of a hold circuit, the continuous-time system and a sampler in cascade, where the signal reconstruction method diagram as shown in Fig 1, and the mathematical formula of FTSH is

$$u_F(t) = \begin{cases} \dfrac{u_Z(t)(kT - t)}{fT} + u_Z(t), t \in [kT, kT + fT), \\ \quad 0, \quad t \in [kT + fT, kT + T), \end{cases} \tag{4}$$

where $k \in N$, $N$ denotes the set of natural, $f \in (0, 1]$ represents the duty cycle of the switched input, and $u_Z(t)$ denotes zero-order hold (ZOH) value in each sampling interval $[kT, kT + T)$. Thus, one can obtain the relation.

$$u_Z(t) = u_Z(kT) = u(kT); \quad kT \leqslant t < kT + T. \tag{5}$$

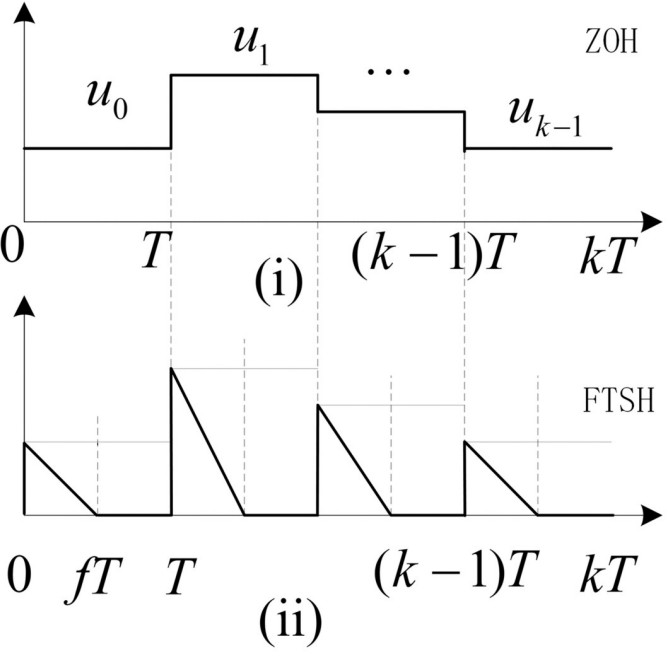

**Fig 1. The diagram of signal reconstruction method ZOH and FTSH [21, 22].**

Then, the state space of corresponding discrete-time system with FTSH is

$$
\begin{cases}
x\left((k+1)T\right) = \Phi_d\, x\left(kT\right) + \Psi u(kT), \\
y\left((k+1)T\right) = c x\left(kT\right),
\end{cases}
\tag{6}
$$

where

$$
\Phi_d = e^{A_c T}, \quad \Psi = \int_0^{fT} e^{A_c(T-\tau)}\left(1 - \frac{\tau}{fT}\right) b d_t.
\tag{7}
$$

Analogously, we can use the following equation to get the discrete-time domain transfer function $G_F(z)$.

$$
\begin{aligned}
G_F(z) &= c(zI - \Phi_d)^{-1}\Psi \\
&= \frac{N_F(z)}{D_F(z)},
\end{aligned}
\tag{8}
$$

the denominator polynomials $D_F(z)$ and numerator polynomial $N_F(z)$ can be deduced by the following equation.

$$
D_F(z) = \det\left(zI - \Phi_d\right),
\tag{9}
$$

$$
N_F(z) = \det\begin{bmatrix} zI - \Phi_d & -\Psi \\ c & 0 \end{bmatrix}.
\tag{10}
$$

Obviously, zeros and poles of discrete-time system $G_F(z)$ could be obtained from (8) by calculating the denominator and numerator polynomials.

In this section, we have reviewed some preliminaries knowledge and previous results of two types systems with FTSH as sample and hold function. The zeros of the discrete-time system, for the case of sufficiently small or large sampling period, are often called limiting zeros [4]. Based on the preliminaries knowledge, in the following section, some more details about the limiting zeros of discrete-time system using FTSH as method will be provided, firstly.

## Limiting zeros of the discrete-time system with FTSH

The purpose of this part is mainly to reveal the properties of limiting zeros about discrete-time system in the case of FTSH. Before showing the corresponding results of limiting zeros of discrete-time system, which are generated by using FTSH as the sample and hold function to construct the input signal. An interesting polynomial in (11) need to be introduced.

**Definition 1** [22]: Suppose that $f \in (0, 1]$, $r \in N \geqslant 0$, exist a number $p$ such that $f + p = 1$. A novel polynomial has the following properties

$$B_{F,r}(z,f) = r! \cdot \det Z_{F,r}, \tag{11}$$

where

$$
Z_{F,r} = \begin{bmatrix}
1 & \dfrac{1}{2!} & \cdots & \dfrac{1}{(r-1)!} & \dfrac{-1+(r+1)f+p^{r+1}}{(r+1)!f} \\[2ex]
1-z & 1 & \cdots & \dfrac{1}{(r-2)!} & \dfrac{-1+rf+p^{r}}{r!f} \\[2ex]
\vdots & \ddots & \ddots & \vdots & \vdots \\[2ex]
0 & \cdots & 1-z & 1 & \dfrac{-1+3f+p^{3}}{3!f} \\[2ex]
0 & \cdots & 0 & 1-z & \dfrac{-1+2f+p^{2}}{2!f}
\end{bmatrix}.
$$

**Remark 1**: Considering the mainly results of this paper, the polynomial in Definition 1 plays a crucial role in revealing the properties of limiting zero of discrete-time system with FTSH. More relevant results about the novel polynomial will be presented in the following part of this section.

**Remark 2**: More details about the method to obtain the result in Definition 1 can refer the process of Theorem 1 and 2. Moreover, Definition 1 reveals that the interrelationship between limiting zeros and relative degree when using the novel sample-and-hold method FTSH.

In order to provide more theoretical reference knowledge about Definition 1 for the potential reader of this paper research topic, several properties of the novel polynomials (11) will be established, firstly. The corresponding results mirror the extensive literature available on the properties of the standard Euler-Fröbenius polynomials—one can see [16, 24, 25].

**Theorem 1**: The polynomials (11) satisfy the following properties:

$$(1) \quad B_{F,r}(z,f) = \sum_{j=1}^{r-1} C_r^j (z-1)^{j-1} B_{F,r-j}(z,f) +$$

$$\frac{-1+(r+1)f+p^{r+1}}{(r+1)!f} \cdot (z-1)^{r-1}$$

(2) $B_{F,r}(z,f) = \frac{r!}{T^r} \det D_{F,r}$ where $T \in R^+ > 0$, and

$$
D_{F,r} = \begin{bmatrix}
T & \dfrac{T^2}{2!} & \cdots & \dfrac{T^{r-1}}{(r-1)!} & \dfrac{(-1+(r+1)f+p^{r+1})T^{r+1}}{(r+1)!fT} \\
1-z & T & \cdots & \dfrac{T^{r-2}}{(r-2)!} & \dfrac{(-1+rf+p^r)T^r}{r!fT} \\
\vdots & \ddots & \ddots & \vdots & \vdots \\
0 & \cdots & 1-z & T & \dfrac{(-1+3f+p^3)T^3}{3!fT} \\
0 & \cdots & 0 & 1-z & \dfrac{(-1+2f+p^2)T^2}{2!fT}
\end{bmatrix} .
\tag{12}
$$

**Proof**:

(1): Based on the results in Definition 1, we can rewrite the expression of $Z_{F,r}$ using the following form

$$
Z_{F,r} = \left[
\begin{array}{c|c}
\begin{matrix} 1 \\ \hline 1-z \\ \vdots \\ 0 \\ 0 \end{matrix} &
\begin{matrix} \frac{1}{2!} & \cdots & \frac{1}{(r-1)!} & \frac{-1+(r+1)f+p^{r+1}}{(r+1)!f} \\[6pt] \\ & & Z_{F,r-1} & \\ \\ \end{matrix}
\end{array}
\right] .
$$

Hence, the Schur determinant Lemma [26] used to compute the above determinant, one can obtain

$$
\begin{aligned}
\det Z_{F,r} ={}& \det Z_{F,r-1} \\
& \times \det \left( 1 - \begin{bmatrix} \dfrac{1}{2!} & \cdots & \dfrac{1}{(r-1)!} & \dfrac{-1+(r+1)f+p^{r+1}}{(r+1)!f} \end{bmatrix} \right. \\
& \left. \times Z_{F,r-1}^{-1} \begin{bmatrix} 1-z \\ 0 \\ \vdots \\ 0 \end{bmatrix} \right) .
\end{aligned}
$$

Note that

$$
Z_{F,r-1}^{-1}
\begin{bmatrix}
1-z \\
0 \\
\vdots \\
0
\end{bmatrix}
= \frac{adjZ_{F,r-1}}{\det Z_{F,r-1}} \cdot
\begin{bmatrix}
1-z \\
0 \\
\vdots \\
0
\end{bmatrix}
$$

$$
= \frac{(adjZ_{F,r-1})_{1,j}^{\mathrm{T}}}{\det Z_{F,r-1}} \cdot (1-z),
$$

where $adjZ_{F,r-1}$ is the adjoint-matrix of $Z_{F,r-1}$, i.e., from the theory of matrices, $(adjZ_{F,r-1})_{i,j} = (-1)^{i+j} \cdot Q_{i,j}$ and the $Q_{i,j}$ is the $(i, j)$ minor of the $Z_{F,r-1}$ and $(adjZ_{F,r-1})_{1,j}$ is the first row of $adjZ_{F,r-1}$.

One can obtain

$$
(adj Z_{F,r-1})_{1,j} = (-1)^{1+j} \times
$$

$$
\det
\begin{bmatrix}
1-z & 1 & \frac{1}{2!} & \cdots & \frac{1}{(j-2)!} & \frac{1}{j!} & \frac{1}{(j+1)!} & \cdots & \frac{1}{(r-3)!} \\
0 & 1-z & 1 & \cdots & \frac{1}{(j-3)!} & \frac{1}{(j-1)!} & \frac{1}{j!} & \cdots & \frac{1}{(r-4)!} \\
\vdots & \vdots & \vdots & \ddots & \vdots & \vdots & \vdots & \ddots & \vdots \\
0 & 0 & 0 & \cdots & 1-z & \frac{1}{2!} & \frac{1}{3!} & \cdots & \frac{1}{(r-j-1)!} \\
0 & 0 & 0 & \cdots & 0 & 1 & \frac{1}{2!} & \cdots & \frac{1}{(r-j-2)!} \\
0 & 0 & 0 & \cdots & 0 & 1-z & 1 & \cdots & \frac{1}{(r-j-3)!} \\
\vdots & \vdots & \vdots & \ddots & \vdots & \vdots & \vdots & \ddots & \vdots \\
0 & 0 & 0 & \cdots & 0 & 0 & 0 & \cdots & 1-z
\end{bmatrix}
\Xi, \tag{13}
$$

where

$$
\Xi =
\begin{bmatrix}
\dfrac{-1 + (r-1)f + p^{r-1}}{(r-1)!f} \\[2ex]
\dfrac{-1 + (r-2)f + p^{r-2}}{(r-2)!f} \\[2ex]
\vdots \\[1ex]
\dfrac{-1 + (r-j+1)f + p^{r-j+1}}{(r-j+1)!f} \\[2ex]
\dfrac{-1 + (r-j)f + p^{r-j}}{(r-j)!f} \\[2ex]
\dfrac{-1 + (r-j-1)f + p^{r-j-1}}{(r-j-1)!f} \\[2ex]
\vdots \\[1ex]
\dfrac{-1 + 2f + p^2}{2!f}
\end{bmatrix}
$$

Calculating the determinant and subsequent sub-determinants along with the first column $j - 1$ times, the following results about the cofactor matrix obtained

$$
\begin{aligned}
(adj Z_{F,r-1})_{1,j} &= (-1)^{j+1}(1-z)^{j-1}\det Z_{F,r-j-1} \\
&= (z-1)^{j-1}\det Z_{F,r-j-1}.
\end{aligned}
$$

Ground on the above properties, using the recursive relation of them, one can get the following results

$$
Z_{F,r-1}^{-1}
\begin{bmatrix}
1 - z \\
0 \\
\vdots \\
0
\end{bmatrix}
= \frac{-1}{\det Z_{F,r-1}}
\begin{bmatrix}
(z-1) \cdot \det Z_{F,r-2} \\
(z-1)^2 \cdot \det Z_{F,r-3} \\
\vdots \\
(z-1)^{r-1} \cdot \det Z_{F,0}
\end{bmatrix},
$$

and consequently, we have

$$
\begin{aligned}
\det Z_{F,r} = \quad & \det Z_{F,r-1} + \cdots + \frac{(z-1)^{r-2}}{(r-1)!}\det Z_{F,1} \\
& + \frac{-1+(r+1)f+p^{r+1}}{(r+1)!f} \cdot (z-1)^{r-1}\det Z_{F,0} \\
= \quad & \sum_{j=1}^{r-1} \frac{(z-1)^{r-j-1}}{(r-j)!}\det Z_{F,j} + \frac{-1+(r+1)f+p^{r+1}}{(r+1)!f} \\
& \cdot (z-1)^{r-1}\det Z_{F,0}.
\end{aligned}
\tag{14}
$$

Using the relation (11) in Definition 1 can complete the proof.

(2): Combined with the Definition 1, it is not hard to find that proving the result of this part is equivalent to proving the determinant $\det D_{F,r} = T^r \det Z_{F,r}$. Here, we use the induction method to prove this part.

The detail steps of induction are as follows:

First, we note that $\det D_{F,1} = T^1 \det Z_{F,1}$ is true.

Then, assuming the result of determinant $\det D_{F,i} = T^i \det Z_{F,i}$ holds for $i = 1, \cdots, r$.

Finally, if we want to prove the result, only need to proof $\det D_{F,r+1} = T^{r+1} \det Z_{F,r+1}$ is also true.

Considering the proof process as shown in the above part (1), we can obtain

$$
\begin{aligned}
\det D_{F,r+1} \quad &= \det D_{F,r} \\
&\times \det \left( T - \left[ \dfrac{T^2}{2!} \quad \cdots \quad \dfrac{T^r}{r!} \quad \dfrac{(-1+(r+2)f+p^{r+2})T^{r+2}}{(r+2)!fT} \right] \right. \\
&\qquad\qquad \left. \times D_{F,r}^{-1} \begin{bmatrix} 1-z \\ 0 \\ \vdots \\ 0 \end{bmatrix} \right)
\end{aligned}
\tag{15}
$$

Using the same compute method in the part (1), it is not difficult to get the following equation

$$
D_{F,r}^{-1} \begin{bmatrix} 1-z \\ 0 \\ \vdots \\ 0 \end{bmatrix} = \dfrac{-1}{\det D_{F,r}} \begin{bmatrix} (z-1) \cdot \det D_{F,r-1} \\ (z-1)^2 \cdot \det D_{F,r-2} \\ \vdots \\ (z-1)^r \cdot \det D_{F,0} \end{bmatrix}
$$

Thus, using the above equation replace the corresponding part in (15) and the determinant of $D_{F,r+1}$ as shown in the following

$$
\begin{aligned}
&\det D_{F,r+1} \\
&= \quad T \det D_{F,r} + \cdots + \dfrac{T^r(z-1)^{r-1}}{r!} \det D_{F,1} \\
&\quad + \dfrac{-1+(r+2)f+p^{r+1}}{(r+2)!f} \cdot T^{r+1}(z-1)^{r-1} \det D_{F,0} \\
&= \quad \sum_{j=1}^{r-1} \dfrac{T^j(z-1)^{j-1}}{j!} \det D_{F,r-j+1} + \\
&\quad \dfrac{-1+(r+2)f+p^{r+1}}{(r+2)!f} \cdot T^{r+1}(z-1)^{r-1} \det D_{F,0}
\end{aligned}
\tag{16}
$$

From the conditions of induction hypothesis, we have assumed that $\det D_{F,i} = T^i \det Z_{F,i}$, $i = 1, \cdots, r$. is true, and then, we have

$$
\begin{aligned}
\det D_{F,r+1} &= T^{r+1} \left(
\begin{array}{l}
\displaystyle\sum_{j=1}^{r-1} \frac{T^j(z-1)^{j-1}}{j!} \det Z_{F,r-j+1} \\[2mm]
\displaystyle + \frac{-1 + (r+2)f + p^{r+1}}{(r+2)!f} \cdot \\[2mm]
(z-1)^{r-1} \det Z_{F,0}
\end{array}
\right) \\[2mm]
&= T^{r+1} \det Z_{F,r+1}
\end{aligned}
\tag{17}
$$

where the last equality can be deduced from the relation of (14). Until now, the second part of Theorem 1 was completely proofed.

Thereafter, the properties of limiting zero dynamics of discrete-time system with FTSH in the case of limit sampling period (i.e. $T \to 0$ or $T \to \infty$, respectively.) will be discussed in the remaining part of this section. Firstly, we reveal the limit expression of sampling zero about sampling period when it is sufficiently small.

**Theorem 2** [22]: When a FTSH as the signal reconstruction to construct the input value of a linear system, one can obtain the following two results.

(1) If the controlled system is composed of a $r$-th order integrator with continuous-time transfer function $G_c(s) = 1/s^r$. Obviously, the relative degree is $r$. We can get the corresponding discrete-time state space expression:

$$
x_{k+1} = A_d x_k + B_{FTSH} u_k
\tag{18}
$$

where

$$
x_k = \begin{bmatrix} x_{1,k} & x_{2,k} & \cdots & x_{r,k} \end{bmatrix}^{\mathrm{T}}
$$

$$
A_d = \begin{bmatrix}
1 & T & \cdots & \dfrac{T^{r-1}}{(r-1)!} \\[2mm]
0 & 1 & \cdots & \dfrac{T^{r-2}}{(r-2)!} \\[2mm]
\vdots & & \ddots & \vdots \\[2mm]
0 & 0 & \cdots & 1
\end{bmatrix},
$$

$$
B_{FTSH} = \begin{bmatrix}
\dfrac{(-1 + (r+1)f + p^{r+1})T^{r+1}}{(r+1)!fT} \\[2mm]
\vdots \\[2mm]
\dfrac{(-1 + 3f + p^3)T^3}{3!fT} \\[2mm]
\dfrac{(-1 + 2f + p^2)T^2}{2!fT}
\end{bmatrix}
$$

and the output of the system is $y_k = x_{1,k}$. Then, the corresponding transfer function $G_F(z)$ of

the exact sampled-data system is

$$G_F(z) = \frac{T^r \cdot B_{F,r}(z,f)}{r! \cdot (z-1)^r} \tag{19}$$

where $B_{F,r}(z,f)$ is the novel polynomials.

(2) If the continuous-time linear system with transfer function $G_c(s)$, which is a strictly proper $n$-th order. Then, the limit expression when $T \to 0$ of the corresponding exact sampled-data model can be shown as follows:

$$\frac{K \cdot (z-1)^m B_{F,r}(z,f)}{(n-m)! \cdot (z-1)^n}, \tag{20}$$

where $r = n - m$ is the relative degree.

**Proof**:

Because of that the proof details of case (1) in Theorem 2 has been provided in [22], for the convenience of potential reader to better understand this Theorem, here, a detailed analysis was carried out for the case (2) in Theorem 2.

*the proof process of case (2)*: Following from the method in [4], one can obtain the ideas to proof the result (20). Specifically, suppose that the continuous-time system $G_c(s)$ is a rational strictly proper (i.e. strictly physical realizable) transfer function expressed as

$$G_c(s) = \frac{K(s-z_1)(s-z_2)\cdots(s-z_m)}{(s-p_1)(s-p_2)\cdots(s-p_n)}, K \neq 0$$

with $r = n - m$ is the relative degree, where $z_i$, $i = 1, 2, \cdots, m$. and $p_j$, $j = 1, 2, \cdots n$. represent zeros and poles, respectively. Then, the exact discrete-time system model can be obtained by using inverse Laplace transform and $\mathcal{Z}$-transform,

$$G_{F,r}(z) = \frac{1}{2\pi j} \int_{\gamma T - j\infty}^{\gamma T + j\infty} \left( \frac{e^\omega}{z - e^\omega} \cdot G_c\left(\frac{\omega}{T}\right) \right.$$

$$\left. \cdot \left( \frac{1 - e^{-f\omega}}{\omega}\left(1 - \frac{1}{f\omega}\right) + \frac{e^{-f\omega}}{\omega} \right) \right) d_\omega$$

where

$$\begin{aligned} G_c\left(\frac{\omega}{T}\right) &= \frac{K \cdot (\omega/T)^m (1 - z_1 T/\omega)\cdots(1 - z_m T/\omega)}{(\omega/T)^n (1 - p_1 T/\omega)\cdots(1 - p_m T/\omega)} \\ &= K \cdot \left(\frac{T}{\omega}\right)^r \frac{(1 - z_1 T/\omega)\cdots(1 - z_m T/\omega)}{(1 - p_1 T/\omega)\cdots(1 - p_m T/\omega)} \end{aligned}$$

Therefore, as $T \to 0$, one can characterize the asymptotic model above the equation as follows

$$\lim_{T \to 0} T^{-r} G_{F,r}(z)$$

$$= \frac{1}{2\pi j} \int_{-j\infty}^{j\infty} \left( \frac{e^\omega}{z - e^\omega} \cdot \left(\frac{K}{\omega^r}\right) \right. \tag{21}$$

$$\left. \cdot \left( \frac{1 - e^{-f\omega}}{\omega}\left(1 - \frac{1}{f\omega}\right) + \frac{e^{-f\omega}}{\omega} \right) \right) d_\omega$$

the integration path of (21) has an infinitesimal detour around origin. Based on the results in part (1), the discrete-time system of an $r$-th order integrator system has the following transfer function

$$G_i(z) = \frac{T^r \cdot B_{F,r}(z,f)}{r! \cdot (z-1)^r} \tag{22}$$

From the above deduce process, we also have the following equation

$$G_i(z) = \mathcal{Z}\left\{\mathcal{L}^{-1}\left\{\frac{1}{s^r}G_{FTSH}(s)\right\}|_{t=kT}\right\}$$

$$= \frac{1}{2\pi j}\int_{-j\infty}^{j\infty}\left(\frac{e^{sT}}{z-e^{sT}}\cdot\frac{1}{s^r}\right.$$

$$\left.\cdot\left(\frac{1-e^{-sfT}}{s}\left(1-\frac{1}{sfT}\right)+\frac{e^{-sfT}}{s}\right)\right)d_s$$

Using $s = \omega/T$ as the method of variable substitution again, we can get

$$G_i(z) = \frac{1}{2\pi j}\int_{-j\infty}^{j\infty}\left(\frac{e^{\omega}}{z-e^{\omega}}\cdot\frac{1}{\omega^r}\right.$$

$$\left.\cdot\left(\frac{1-e^{-f\omega}}{\omega}\left(1-\frac{1}{f\omega}\right)+\frac{e^{-f\omega}}{\omega}\right)\right)d_{\omega} \tag{23}$$

Finally, combining (21), (22) and (23), one can use the following limit expression to characterize the asymptotic model

$$\lim_{T\to 0}T^{-r}G_{F,r}(z) = \frac{K\cdot B_{F,r}(z,f)}{(n-m)!\cdot(z-1)^r}$$

**Remark 3**: In Theorem 2, we notice that for the general strictly proper linear system (1) when the signal is reconstructed by FTSH. It is easy to know that $n$ poles converge to 1 as $e^{p_i T}$ of the corresponding discrete-time system when the sampling period $T \to 0$, and $m$ zeros close to 1, moreover, other zeros are equivalent to the roots of $B_{F,r}(z,f)$. Obviously, the sampling zeros are influenced by the values of relative degree and the duty cycle value $f$ of FTSH.

Next, a result of limiting zeros for a sufficiently large sampling period in the case of FTSH will be deduced in this part of this section. Åström et al. [4] have shown that all the zeros of discrete-time system with ZOH approach $z = 0$ as $T \to \infty$ when the original continuous-time system $G_c(s)$ is stable and satisfy the initial condition $G_c(0) \neq 0$. However, the situation of system exists unstable part is not considered. Hagiwara et al. [6] have continued in-depth to study the case when original continuous-time controlled system has unstable poles and have shown the properties of limiting zeros of sampled-data system with ZOH and FOH, analytically. The corresponding research about limiting zeros in the case of FROH and BTSH has been explored in [9, 18], respectively. In next part of this section, we will explore the capabilities of limiting zero dynamics of discrete-time system with FTSH under large sampling period which will greatly enrich the theoretical framework of discrete-time system with FTSH.

**Theorem 3**: Assuming that $G_c(s)$ in (1) is not having any pole on the imaginary axis. $n_s$ and $n_a (= n - n_s)$ are the number of stable and unstable poles, respectively. Thereafter, suppose that the continuous-time system $G_c(s)$ can be decomposed into the following expression:

$$G_c(s) = G_s(s) + G_a(s) \tag{24}$$

where $G_s(s)$ is the stable part, $G_a(s)$ is unstable part, and

$$G_s(0) \neq 0 \quad \text{or} \quad G_a(0) \neq 0. \tag{25}$$

Then, zeros of discrete-time system $G_F(z)$ with $f \neq 0$ have the following properties when $T \to \infty$:

$$
B_F(z) = B_F\left(\alpha + \frac{1}{x}\right)
$$

$$
= \det \begin{bmatrix}
\left(\alpha + \frac{1}{x}\right)I - e^{A_a T} & 0 & -\int_0^{fT} e^{A_a(T-\tau)}\left(1 - \frac{\tau}{fT}\right)b_a d\tau \\
0 & \left(\alpha + \frac{1}{x}\right)I - e^{A_s T} & -\int_0^{fT} e^{A_s(T-\tau)}\left(1 - \frac{\tau}{fT}\right)b_s d\tau \\
c_a & c_s & 0
\end{bmatrix}
$$

$$
= \frac{1}{x^{n-1}} \det e^{A_a T} \times \tag{26}
$$

$$
\left\{ \det\left((\alpha x + 1)I - x e^{A_s T}\right) \det \begin{bmatrix}
(\alpha x + 1)e^{-A_a T} - xI & -\int_0^{fT} e^{-A_a \tau}\left(1 - \frac{\tau}{fT}\right)b_a d\tau \\
c_a & 0
\end{bmatrix} \right.
$$

$$
\left. + \det\left((\alpha x + 1)e^{-A_a T} - xI\right) \det \begin{bmatrix}
(\alpha x + 1)I - x e^{A_s T} & -\int_0^{fT} e^{A_s(T-\tau)}\left(1 - \frac{\tau}{fT}\right)b_s d\tau \\
c_s & 0
\end{bmatrix} \right\}
$$

*Case (i) $n_a = 0$* (i.e. continuous-time system $G_c(s)$ is stable): All the zeros of $G_F(z)$ converge to $z = 0$ as $T \to \infty$.

*Case (ii) $1 \leqslant n_a \leqslant n - 1$*: If $G_a(0) \neq 0$, then $n_s$ zeros of $G_F(z)$ go to the origin, and remaining $n_a - 1$ zeros diverge to infinity as the sampling period $T \to \infty$.

*Case (iii) $n_a = n$*: All zeros of the final discrete-time system $G_F(z)$ close to infinity large as $T \to \infty$.

**Proof**: The process of proof is the same as [6, 9]. Considering the structure of results in three cases are similar. Here, we provide the proof process of *case (ii)* in this Theorem. The proof details about *case (i)* and *case (iii)* are omitted because only need a small changes that can be obtained from the proof process of *case (ii)*.

Based on the assumption, the state space matrix of $G_c(s)$ can rewrite as the following form

$$
A_c = \begin{bmatrix} A_a & 0 \\ 0 & A_s \end{bmatrix}, b = \begin{bmatrix} b_a \\ b_s \end{bmatrix} \text{ and } c = [\, c_a \quad c_s \,].
$$

Thus $G_s(s) = c_s(sI - A_s)^{-1}b_s$, $G_a(s) = c_a(sI - A_a)^{-1}b_a$.

Let

$$z = \alpha + \frac{1}{x} \tag{27}$$

where $\alpha$ represents the real number and it is satisfies

$$\alpha \neq 0, \ G_a(0) \neq 0 \tag{28}$$

Then, based on the decomposed realization form and (10), one can obtain (26) (as shown in the top pf this page)

Using the same steps as shown in [6],

$$B\left(\alpha + \frac{1}{x}\right)$$

$$\rightarrow \frac{1}{x^{n-1}} \det e^{A_a T} (-1)^{n_a-1} x^{n_a-1} (\alpha x + 1)^{n_s} c_a A_a^{-1} b_a$$

$$= \frac{1}{x^{n-1}} \det e^{A_a T} (-1)^{n_a-1} x^{n_a-1} (\alpha x + 1)^{n_s} G_a(0)$$

as $T \rightarrow \infty$. Thus, if $G_a(0) \neq 0$, the sampled-data system has $n_a-1$ roots diverge to $x = 0$, $n_s$ roots approach to $x = -\frac{1}{\alpha}$. Based on the relation of them in (27), one can obtained the results in case (ii). And the proof is complete.

## Criteria for the limiting zero stabilization

In the above section, we have provided the limiting expression of sampling zeros of discrete-time system with limit sampling period. But we only gave the expression about the sampling zeros. Therefore, to reveal the stability of sampling zeros is an important work. Thereafter, the stable conditions of limiting zeros of discrete-time system with FTSH need to be researched and provided precisely. Firstly, the stability condition of limiting zeros of sampled-data system in the case of sufficiently small sampling period is shown using the following theorem.

**Theorem 4** [22]: Assuming that the original continuous-time system $G_c(s)$ has no zeros on the imaginary axis. And its corresponding discrete-time system is generated by using FTSH in the case of sufficiently small sampling period. All the limiting zeros are stable when $r = 2$ if all the zeros of $G_c(s)$ are stable and the selectable variable parameter $f$ satisfies $0 < f \leqslant 1$.

**Remark 4**: Generally speaking, the relative degree of many linear or nonlinear mechanical systems in practical field is two [27, 28]. As the results in Theorem 4 shown, one can find the feasible solution of the variable parameter of FTSH to ensure the corresponding discrete-time limiting zeros are stable.

**Remark 5**: An insightful observation in the Theorem 4 is that the stability properties and conditions are not given when the relative degree of continuous-time system is greater than two. Using Jury's test can get the stability conditions of the limiting zeros by computing the roots of corresponding numerator polynomial $B_{F,r}(z, f) = 0$. Thus, we notice that it cannot find a proper $f$ to let limiting zeros stable when the relative degree of continuous-time system is three.

Next, the stable condition of limiting zeros of discrete-time system in the case of sufficiently large sampling period is also provided using the following theorem.

**Theorem 5**: Suppose that original continuous-time system $G_c(s)$ has no poles on the imaginary axis, and let the number of its unstable poles is $n_a$. Further assume that $G_c(s)$ is decomposed into (24). we have

*Case(i) $n_a = 0$*: The system $G_c(s)$ is stable, when the sampling period $T$ is sufficiently large, if want to make the zeros of $G_F(z)$ are stable, the controlled system should satisfy

$$G_c(0) \neq 0 \tag{29}$$

*Case(ii) $n_a = 1$*: When using large sampling period $T$, all the discrete-time zeros of $G_F(z)$ are stable and converge to zero if and only if

$$G_a(0) \neq 0 \tag{30}$$

*Case(iii) $n_a \geqslant 2$*: The corresponding sampled discrete-time system $G_F(z)$ always has an unstable limiting zeros when the sampling period is sufficiently large.

**Proof**: Based on the previous analysis and proof process in Theorem 3, the corresponding results in this Theorem can be deduced easily. Thus, the proof details of this Theorem are omitted.

**Remark 6**: Comparing the results of this paper and the corresponding results with BTSH in [9], when the sampling period is sufficiently large, the FTSH has better performance for limiting zeros than BTSH. For example, the resulting discrete-time system has minimum phase (MP) characteristics, using FTSH as signal reconstruction method, when the original continuous-time systems have an unstable pole. However, the BTSH cannot provide this property.

**Remark 7**: Compared with the results of limiting zeros between FTSH and BTSH for sufficiently small sampling period. The feasible region (or interval) of the FTSH is larger than BTSH.

In the previous research, it is shown that FTSH has the outstanding advantage compared with the BTSH with selecting the suitable sample-and-hold parameters based on the simulation studies [21]. And in the mainly part of this paper, it has analytically revealed that FTSH can provide a better performance of limiting zeros and a larger feasible region than BTSH. Furthermore, the advantages of FTSH will be validated by some numerical simulations in the next section.

## Numerical simulations

From the above analyses we can know that the properties of limiting zeros for discrete-time system model with FTSH are connected with the relative degree, sampling period and the number of unstable poles. Thereby, we provide some interesting numerical examples to verify the results in the main part of this study.

Firstly, the continuous-time system with relative degree two and its' continuous-time transfer function is

$$G(s) = \frac{s+7}{(s+1)(s+2)(s+3)}. \tag{31}$$

Obviously, the original controlled continuous-time system is stable and minimum phase. Based on the corresponding results in Theorem 2 and Theorem 4, we can select the appropriate parameter $f$ of FTSH to replace the limiting zero and let it is stable. Therefore, we choose the parameter of FTSH and the location of limiting zeros of (31) simulation result is shown in Fig 2 (This numerical can also refer [22]. But in our previous research work, system with unstable part is not mentioned, which will be analysized in the next part.). As shown in simulation diagram in Fig 2, the discrete-time zeros can be placed inside the unite circle no matter that the sampling period is small or large. Moreover, the phenomenon in simulation are coinciding with the results in Theorem 2 and 3.

Next, we need to consider the situation that the original controlled system has unstable real poles. Combining with the results in Theorem 3 and Theorem 5, two examples with different number unstable real poles are given to verify the results. First of all, one can select a

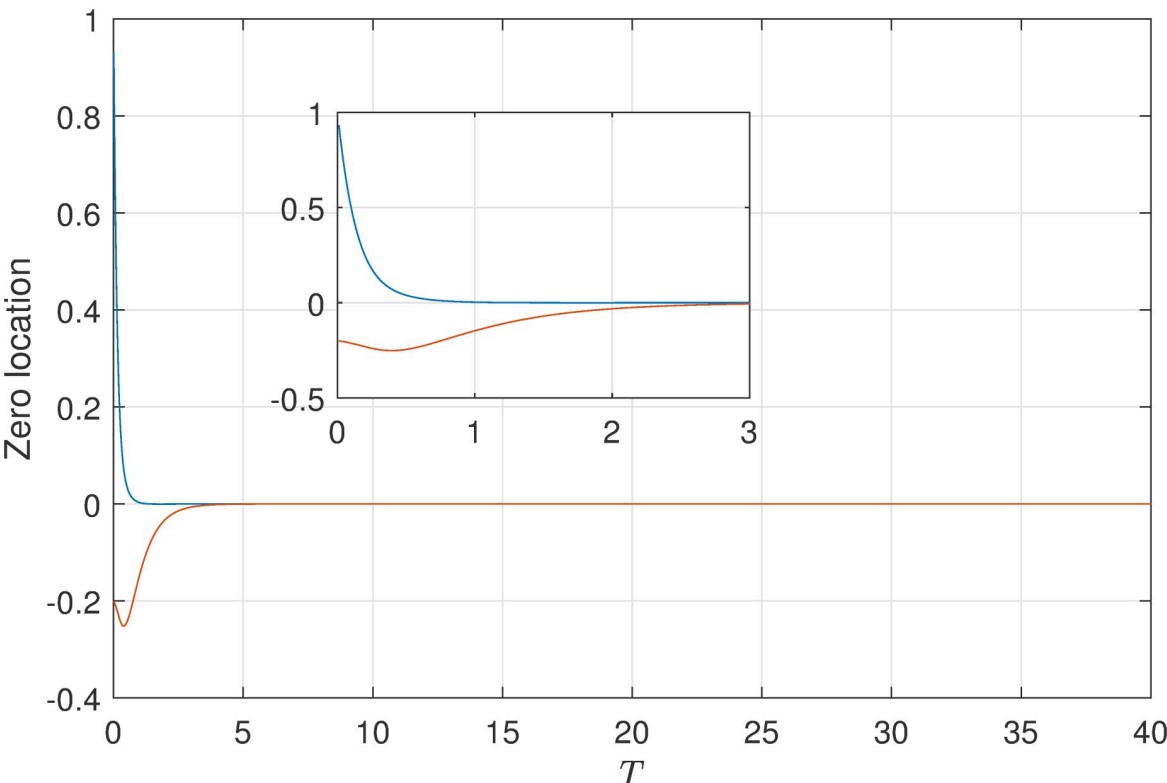

**Fig 2. The variation tendency of limiting zeros for system (31) with respect to $T$ when the parameter $f = 0.5$ of FTSH [22].**

continuous-time system with one unstable real pole as follows

$$G_{n_a=1}(s) = \frac{s+7}{(s-1)(s+2)(s+3)}, \tag{32}$$

where $n_s = 2$ and $n_a = 1$. The results in Theorem 3 and Theorem 5 indicate that the limiting zeros converge to 0 as $T \rightarrow \infty$ (i.e. $1/T \rightarrow 0$). Obviously, the numerical simulation verified the corresponding results in this study are correct, and the diagram as showed in Fig 3.

Another side, we consider a system with two unstable real poles, assume the transfer function is

$$G_{n_a=2}(s) = \frac{s+7}{(s-1)(s-2)(s+3)}, \tag{33}$$

where $n_s = 1$ and $n_a = 2$.

According to the theoretical results in Theorem 3 and 5, for the sufficiently large sampling period $T$, the zeros of the corresponding discrete-time system have two classes: one approach to infinity, another one diverges to zero. Eventually, the simulation results about system (33) are shown in Fig 4. Obviously, the simulation results and the theoretical results are coincide.

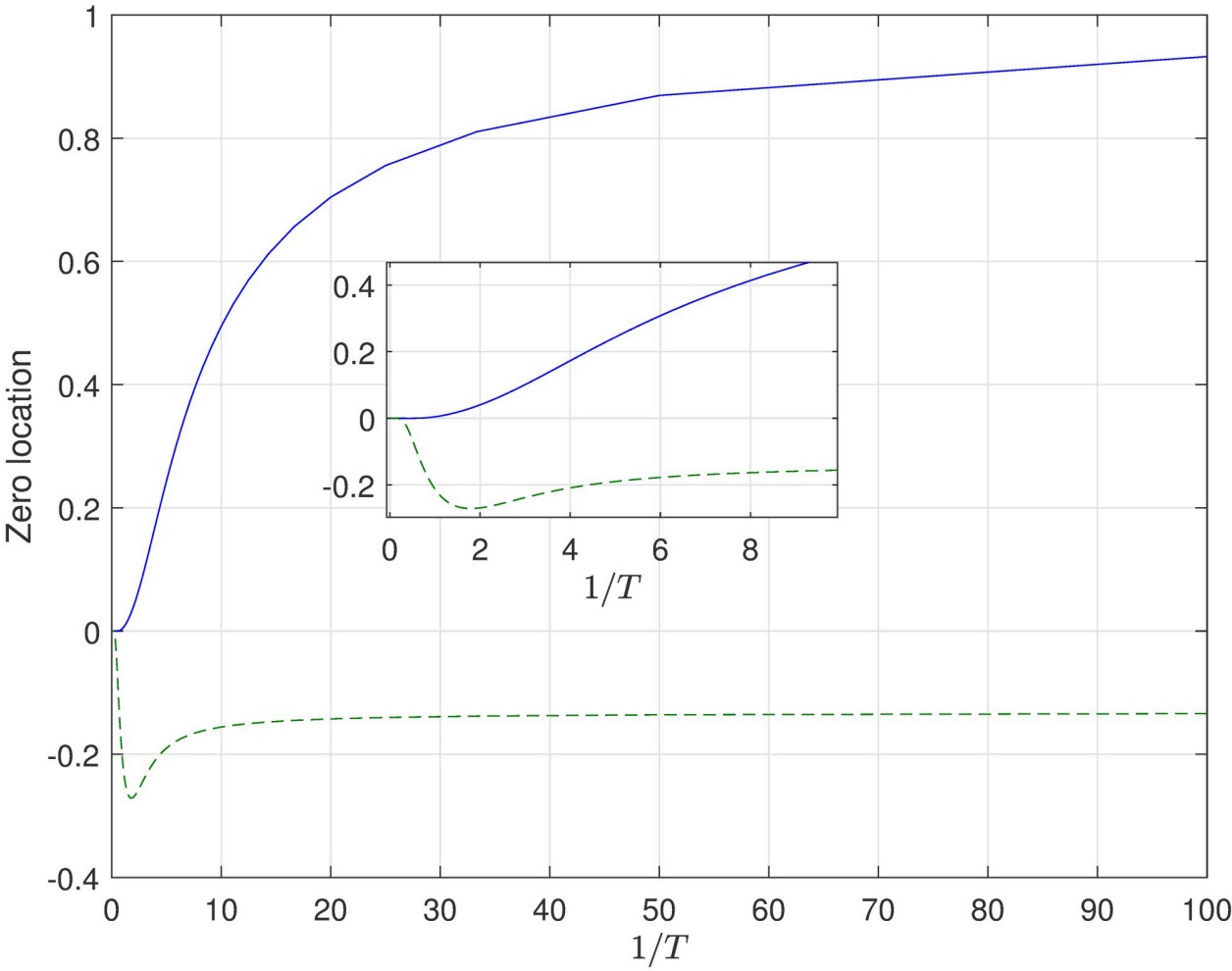

**Fig 3. The location of discrete-time zeros for system with one unstable real pole in the case of FTSH and parameter $f = 0.35$.**

Finally, in order to confirm the result that FTSH has the outstanding advantage compared with the BTSH, which need to illustrate by a numerical simulation, we also introduce the Disk drives serves (see Fig 5) [9, 29] as a didactic practical example to verify the advantages.

We select parameters' value of Disk drives are same as in [9, 29]. Then, the transfer function of Disk read-write head model is the relative degree two and has the following form

$$G(s) = \frac{5}{(s + 0.2)^2 + 31.62}. \tag{34}$$

Combined with the feasible interval of sample and hold parameter, we select the device parameter of BTSH and FTSH with $f = 0.55$ and $T \in (0, 0.2]$.

Then, the limiting zeros of the corresponding sampled-data system with two sample and hold devices (i.e. FTSH and BTSH) are presented in Fig 6. Evidently, the limiting zeros always can be replaced in the unite circle region by selecting the value of BTSH and FTSH. And interestingly, the value of limiting zeros in the case of BTSH is larger than FTSH (i.e. the location of sampling zeros in the case of FTSH are closer to origin than BTSH).

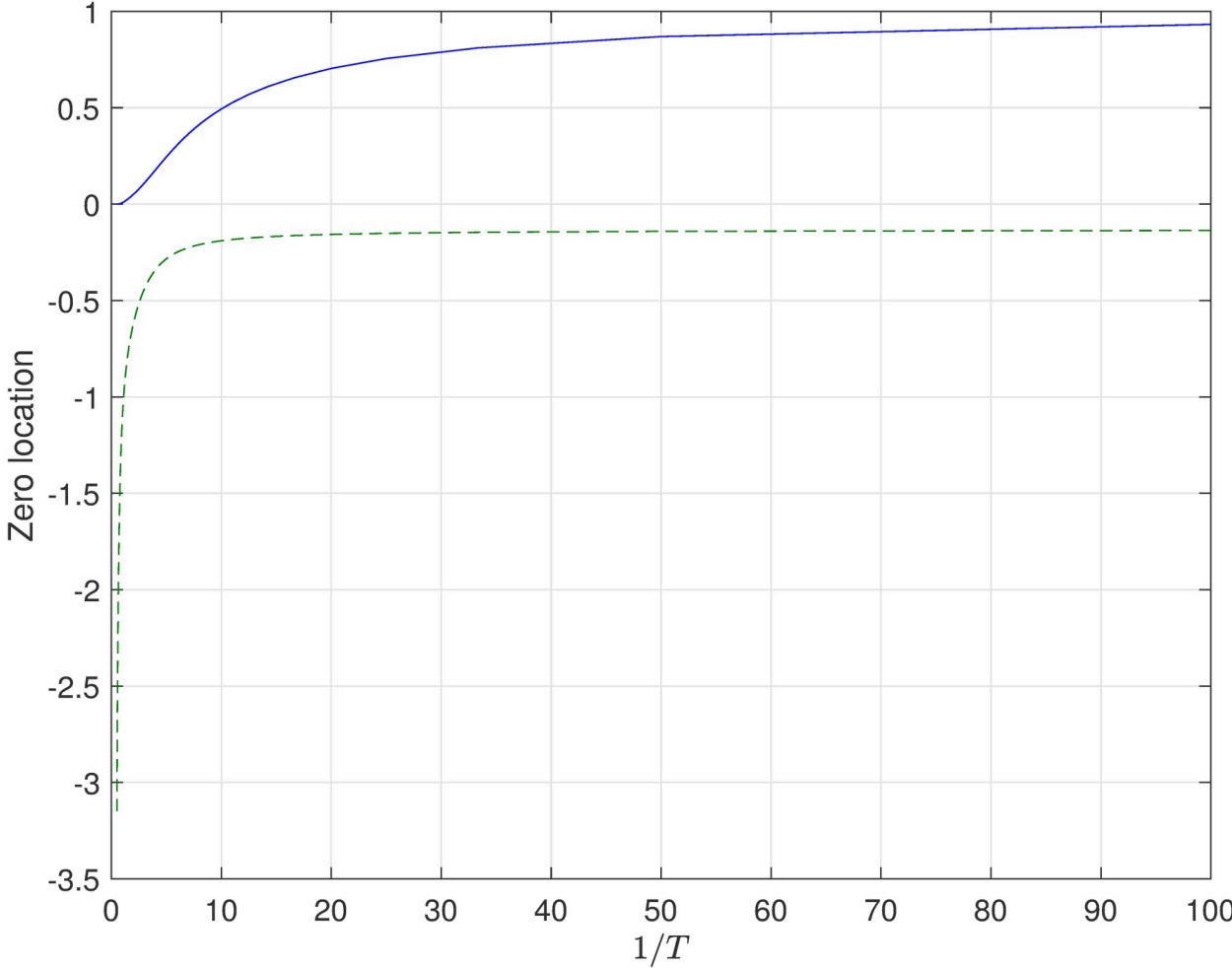

**Fig 4. The location of discrete-time zeros for system with two unstable real poles in the case of FTSH and parameter $f = 0.35$.**

Furthermore, we also select the traditional ZOH as the alternative signal reconstruction method to generate the inputs of the Disk drivers system (34). The absolute values of limiting zeros of the system as shown in Fig 7.

Obviously, we can not to let the corresponding discrete-time system is MP by selecting a proper sampling period when use the traditional ZOH as the signal reconstruction method (i.e. the discrete-time system always has a zero locate outside the unite circle.).

In Fig 6, we only have shown the location of sampling zeros with respect to sampling period $T$ under a fixed duty cycle $f$. However, it is not clear to know the location of sampling zeros with different $f$. Then, one can select a fixed sampling period $T = 0.1$, and show the comparative simulations of BTSH and FTSH about the location of sampling zeros for system (34) with different duty cycle $f$ in Fig 8. Based on the previous research in [9], the feasible interval of $f$ is (0, 0.75] in the case of BTSH. As shown in Fig 8, the simulation result of BTSH is coincide with [9]. Another side, the sampling zero locates on the boundary of unite circle when $f = 0.75$. Both in the case of BTSH and FTSH, the simulation results and the theoretical results are coincide.

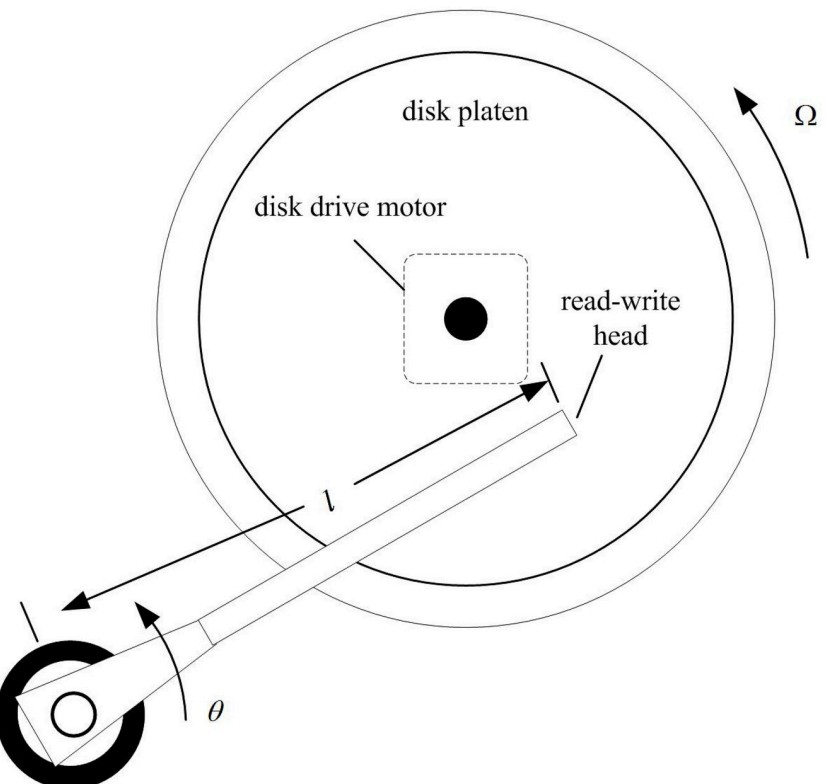

**Fig 5. The diagram of computer hard disk drive [9].**

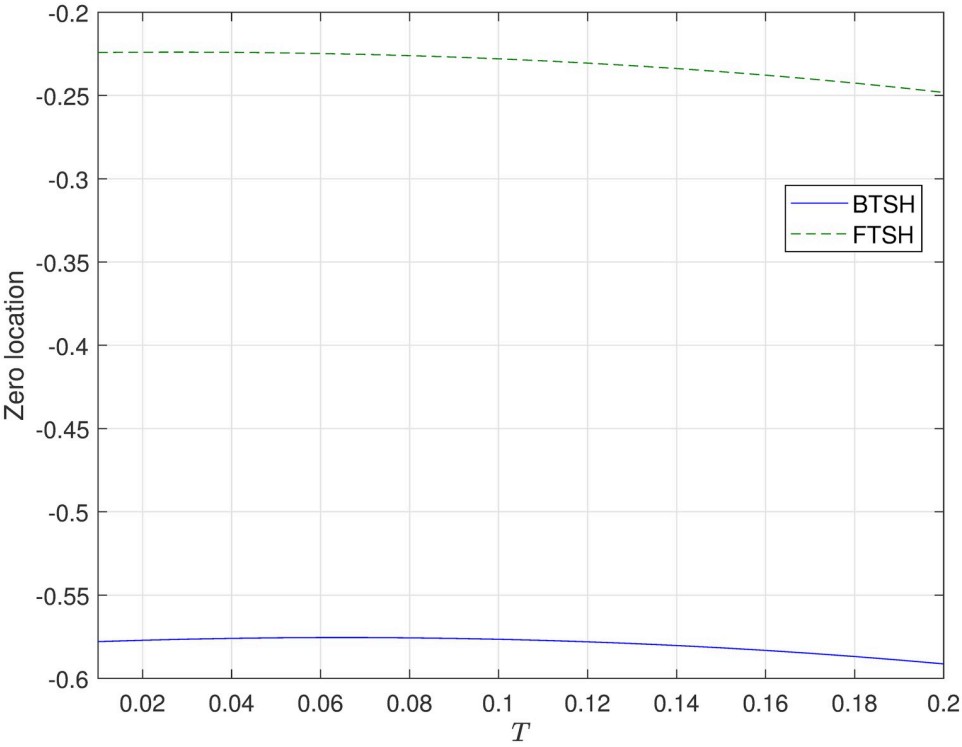

**Fig 6. The location of discrete-time zeros of the disk read-write head system (34) with FTSH, BTSH.**

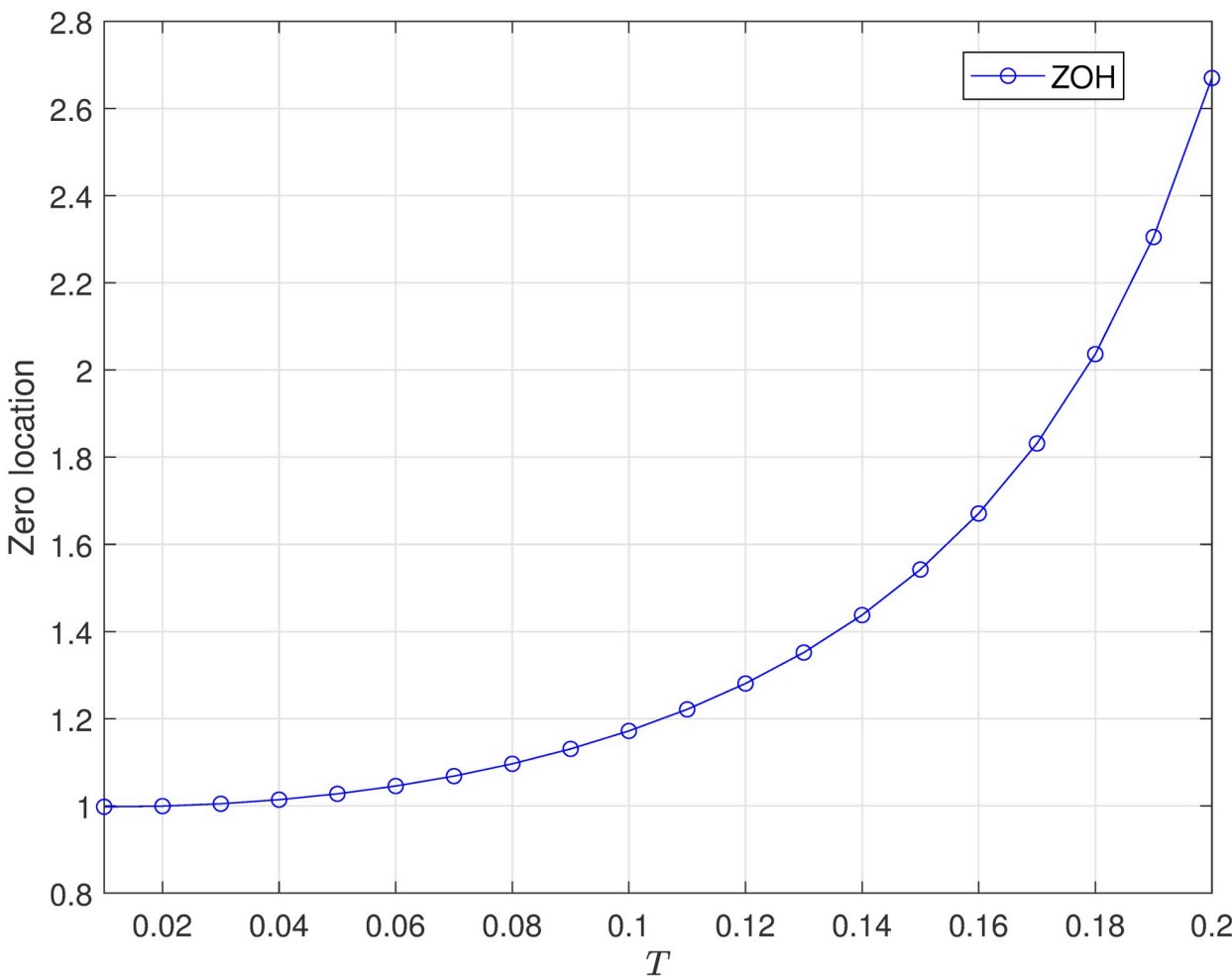

**Fig 7. The absolute values of discrete-time zero of disk read-write head system (34) with ZOH.**

## Conclusion

This paper has dealt with the properties of limiting zero dynamics of sampled-data system with FTSH as the sample and hold device. The expression of limiting zero polynomial of continuous-time linear system in the case of FTSH has been given. And the stable condition of sampling zeros of the corresponding system has also been provided for the case of sufficiently small sampling period, where FTSH can replace the sampling zeros of the sampled-data system into the stable region while ZOH fails to do so. Moreover, this paper has also derived the properties of sampling zeros of sampled-data system with FTSH for sufficiently large sampling period, FTSH can let all zeros of the discrete-time systems are stable when the original continuous-time system has one unstable poles while BTSH can not provide this properties. Compared with the previous results, this paper was analytical extending work from the simulation research in [21]. It has also analytically revealed that FTSH can provide a better performance of limiting zeros and a larger feasible region than BTSH. We have verified the results in this

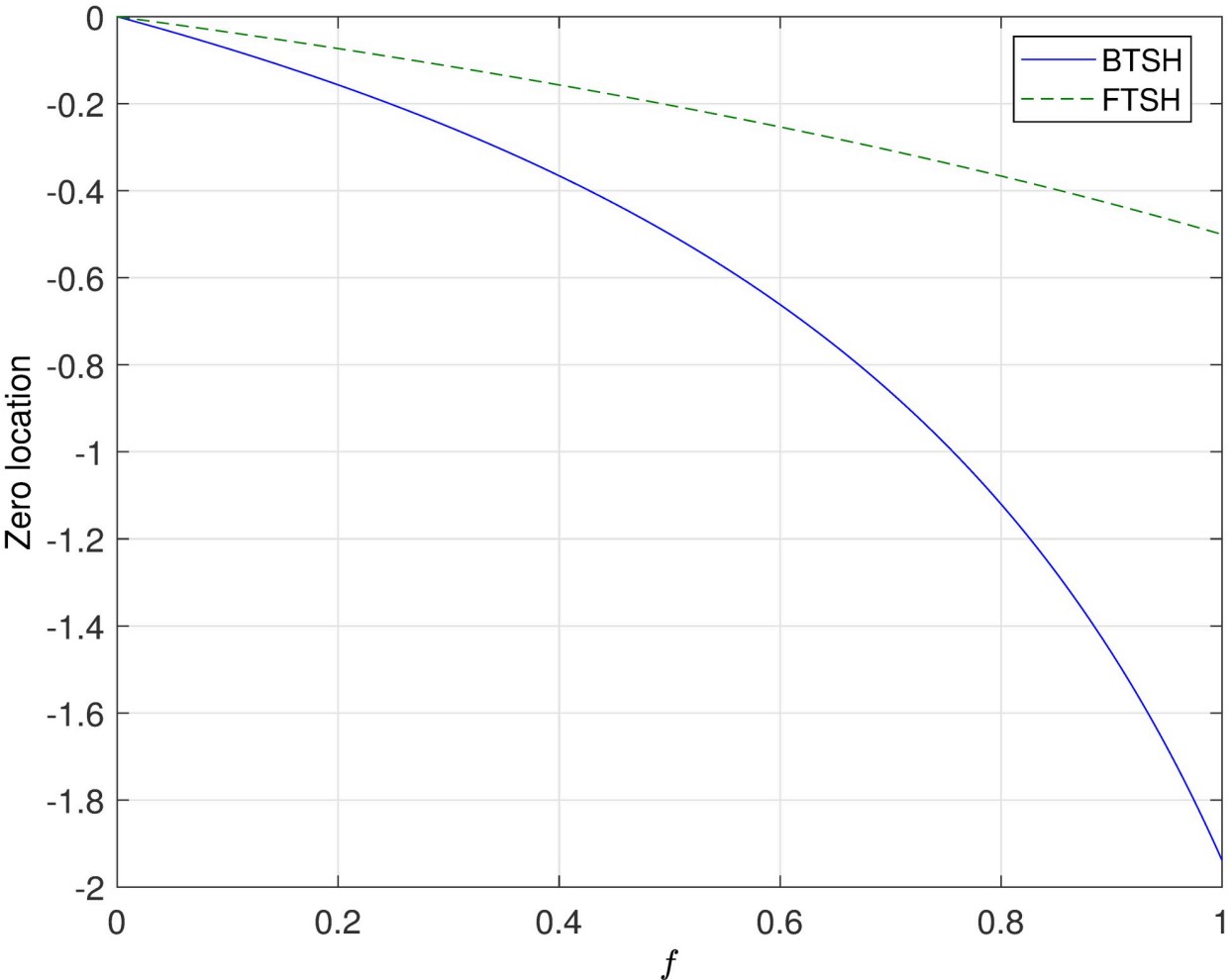

**Fig 8. Convergence of sampling zeros of the disk read-write head system (34) with no fixed duty cycle *f*.**

paper are correct through numerical simulation. Further work will concentrate on the characteristics of zero dynamics for nonlinear system with FTSH.

## Author Contributions

**Conceptualization:** Minghui Ou, Haiyang Wang.

**Data curation:** Minghui Ou, Haiyang Wang.

**Formal analysis:** Minghui Ou.

**Funding acquisition:** Minghui Ou, Haiyang Wang.

**Investigation:** Minghui Ou.

**Methodology:** Minghui Ou.

**Project administration:** Minghui Ou.

**Resources:** Minghui Ou.

**Software:** Minghui Ou, Haiyang Wang.

**Supervision:** Minghui Ou, Mingkun Ou.

**Validation:** Minghui Ou.

**Visualization:** Minghui Ou, Haiyang Wang.

**Writing – original draft:** Minghui Ou.

**Writing – review & editing:** Minghui Ou, Mingkun Ou, Haiyang Wang.

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
