## [Decision Letter · Decision Letter 0]

13 Jan 2023

PONE-D-22-34561The limiting zero dynamics of discrete-time system based on forward triangle sample-and-holdPLOS ONE

Dear Dr. ou,

Thank you for submitting your manuscript to PLOS ONE. After careful consideration, we feel that it has merit but does not fully meet PLOS ONE’s publication criteria as it currently stands. Therefore, we invite you to submit a revised version of the manuscript that addresses the points raised during the review process.

We look forward to receiving your revised manuscript.

Kind regards,

Junyuan Yang

Academic Editor

PLOS ONE

“This research was funded by the Science and Technology Research Program of Chongqing Municipal Education Commission ( Grant No. KJZD-M202203401, KJQN202103401, KJQN202103413) and the Natural Science Foundation of Chongqing, China, (Grant No. cstc2021jcyj-msxmX0532, cstc2021ycjh-bgzxm0088), Program for Innovation Research Groups at Institutions of Higher Education in Chongqing (Grant No. CXQT21032), the University's Scientific Research Program of Chongqing Vocational Institute of Engineering (Grant No. 2022KJA03, JG222024).”

Reviewers' comments:

Reviewer's Responses to Questions

**Comments to the Author**

1. Is the manuscript technically sound, and do the data support the conclusions?

Reviewer #1: Yes

Reviewer #2: Yes

2. Has the statistical analysis been performed appropriately and rigorously? 

Reviewer #1: Yes

Reviewer #2: Yes

3. Have the authors made all data underlying the findings in their manuscript fully available?

Reviewer #1: Yes

Reviewer #2: Yes

4. Is the manuscript presented in an intelligible fashion and written in standard English?

Reviewer #1: No

Reviewer #2: Yes

5. Review Comments to the Author

Reviewer #1: The review on “The limiting zero dynamics of discrete-time system based on

forward triangle sample-and-hold”

Authors: Minghui Ou, Haiyang Wang

This paper systematically reveals the properties of limiting zeros of discrete-time system with forward triangle sample-and-hold(FTSH) in details. Further, it presents the stable conditions of the limiting zeros. This paper analytically reveals the truth that FTSH has the outstanding advantage compared with the BTSH. The theoretical framework of properties of limiting zeros for some eligible systems with FTSH was provided. However, for the better version of this manuscript, I propose some questions and suggestions as follows:

1. Page 3, line 89: “compute” should corrected as “computing”;

2. Page 4, line 188: Is this a Lemma? The equation (11) more like a definition rather than a property.

3. Page 5, line 136: Is there a corresponding references about” Schur determinant Lemma”? Please add it.

4. Page 10, line 195 and line 197: “There have ... “ They’re incorrect sentence patterns.

5. Page 10 line 203: Delete “should”;

6. Page 10 line 205: The verb tense of “continue” is wrong;

7. Page 14 line 359: Please write down the correct picture number?

8. Fig 1: Please give a clear picture

Reviewer #2: This study concerns with the properties of limiting zero dynamics of the resulting discrete-time systems. The results of the limiting zero dynamics in the situation of sufficiently small or large sampled period is introduced. Then, the stable conditions of the limiting zeros is presented.

The topic of this paper is interesting. The methods of this paper is novel. The organizations of this paper and the written style look well. However, a number of points need to be clarified and certain statements require further justification.

6. PLOS authors have the option to publish the peer review history of their article (what does this mean?). If published, this will include your full peer review and any attached files.

Reviewer #1: No

Reviewer #2: No

---

## [Author Response · Author response to Decision Letter 0]

15 Feb 2023

Reviewer 1:

Q1. Page 3, line 89: “compute” should corrected as “computing”;

R1. Thanks for your helpful suggestion. We have carefully revised the language and grammar problems in the revised manuscript.

Q2. Page 4, line 188: Is this a Lemma? The equation (11) more like a definition rather than a property.

R2. Thanks very much for your useful advice. The original manuscript is Lemma, which is not precise. In the revised version, we have correctted it as Definition.

Q3. Page 5, line 136: Is there a corresponding references about” Schur determinant Lemma”? Please add it.

R3. Thanks very much for your kindly remind. We have added the corresponding reference paper about the Schur determinant Lemma in the revised version. 

Q4. Page 10, line 195 and line 197: “There have ... “ They’re incorrect sentence patterns.

R4. Thanks very much for your beneficial suggestion. We have carefully revised the language and grammar problems in the revised manuscript.

Q5. Page 10 line 203: Delete “should”;

R5. Thanks very much for your useful suggestion. We have deleted “should” in the revised version.

Q6. Page 10 line 205: The verb tense of “continue” is wrong;

R6. Thanks very much for your useful advice. We have revisied verb tense problem.

Q7. Page 14 line 359: Please write down the correct picture number?

R7. Thanks very much for your useful suggestion. We have writed the correct picture number in the revised version.

Q8. Fig 1: Please give a clear picture

R8. Thanks very much for your kindly remind. The clear picture of Fig 1 was corrected. And the new picture as shown as follows.

Reviewer 2:

Q1. The setting of the figures is not standard. For example, the size and font of the text in the figure, the thickness of the lines, etc.

R1. Thanks very much for your very helpful suggestion. Firstly, the fig 1 is not clear, we have provided a more clearly picture in the revised version manuscript. Secondly, we have changed the problem about the text of figure and thickness of the lines in other figures.

Q2. I would encourage the author to extend the abstract more with the key results, and the description of highlights and main conclusions should also be improved at the end.

R2. Thanks very much for your useful advice. In abstract, we introduce the key results about the framework of limiting zero dynamics and the corresponding stable conditions in sufficiently small and large sample period, and also provide the feasible region about the variable of FTSH to replace the sampling zeros locate inside the stable region. In conclusion, we have rewritten this part. In the revised version, we firstly introduced the research topic of this paper, and we compared with the previous research results to summarize this paper results. The corresponding improved change can see in revised version. 

Q3. On page 14, line 359, the writing is not standardized, “As shown in Fig.??”.

R3. Thanks very much for your helpful suggestion. We have changed the description in the revised version.

Q4. The format of the references is not uniform, e.g. 24.

R4. Thanks very much for your useful suggestion. We have carefully check the reference format. Because of the references 24 and 25 is a book, and we use the PLOS ONE template “plos_latex_template.tex” to construct the manuscript. In the revised version, the format of journal references is uniform and the format of book references is uniform.

---

## [Editor Report · Decision Letter 1]

2 Mar 2023

The limiting zero dynamics of discrete-time system based on forward triangle sample-and-hold

PONE-D-22-34561R1

Dear Dr. Ou,

We’re pleased to inform you that your manuscript has been judged scientifically suitable for publication and will be formally accepted for publication once it meets all outstanding technical requirements.

Kind regards,

Junyuan Yang

Academic Editor

PLOS ONE
---

## [Editor Report · Acceptance letter]

31 Mar 2023

PONE-D-22-34561R1 

The limiting zero dynamics of discrete-time system based on forward triangle sample-and-hold 

Dear Dr. Ou:

I'm pleased to inform you that your manuscript has been deemed suitable for publication in PLOS ONE. Congratulations! Your manuscript is now with our production department. 

Kind regards, 

on behalf of

Dr. Junyuan Yang 

Academic Editor

PLOS ONE